



# Mechanisms of hydrological responses to volcanic eruptions in the Asian monsoon and westerlies-dominated subregions

Zhihong Zhuo[1,a], Ingo Kirchner[1], and Ulrich Cubasch[1]

[1]1Institute of Meteorology, Freie Universität Berlin, 12165 Berlin, Germany
[a]now at: Section for Meteorology and Oceanography, Department of Geosciences, University of Oslo, 0315 Oslo, Norway

**Correspondence:** Zhihong Zhuo (zhihong.zhuo@met.fu-berlin.de)

**Abstract.** Explosive volcanic eruptions affect surface climate especially in monsoon regions, but responses vary in different regions and to volcanic aerosol injection (VAI) in different hemispheres. Here we use six ensemble members from last millennium experiment of the Coupled Model Intercomparison Project Phase 5, to investigate the mechanism of regional hydrological responses to different hemispheric VAI in the Asian monsoon region (AMR). It brings a significant drying effect over the AMR after northern hemisphere VAI (NHVAI), spatially, a distinct "wet get drier, dry gets wetter" response pattern emerges with significant drying effect in the wettest area (RWA) but significant wetting effect in the driest area (RDA) of the AMR. After southern hemisphere VAI (SHVAI), it shows a significant wetting effect over the AMR, but spatial response pattern is not that clear due to small aerosol magnitude. The mechanism of the hydrological impact relates to the indirect change of atmospheric circulation due to the direct radiative effect of volcanic aerosols. The decreased thermal contrast between the land and the ocean after NHVAI results in weakened EASM and SASM. This changes the moisture transport and cloud formation in the monsoon and westerlies-dominated subregions. The subsequent radiative effect and physical feedbacks of local clouds lead to different drying and wetting effects in different areas. Results here indicate that future volcanic eruptions may alleviate the uneven distribution of precipitation in the AMR, which should be considered in the near-term decadal prediction and future strategy of local adaptation to global warming. The local hydrological responses and mechanisms found here can also provide reference to stratospheric aerosol engineering.

## 1 Introduction

The Asian monsoon region (AMR, 8.75°S–56.25°N, 61.25°CE–143.75°E, Cook et al., 2010) is the most densely-populated region all over the world. As part of the largest continental landmass, the climate here shows large regional differences. Figure 1 shows the dominant climate systems and climatological precipitation distribution in the boreal summer (June-July-August, JJA) in 1981-2010. The purple line indicates the modern Asian summer monsoon limit, to the northwest are the westerlies-dominated arid areas, to the southeast are the monsoon-dominated humid areas, due to the contrast between the landmass and the Indian and Pacific Ocean (Dando, 2005; Chen et al., 2008). It includes two monsoon subsystems – the East Asian summer monsoon (EASM) and the South Asian summer monsoon (SASM), which are usually separated by 100°E longitude (Chiang et al., 2017). The precipitation is unevenly distributed with a diminishing scale from southeast to northwest in the AMR.





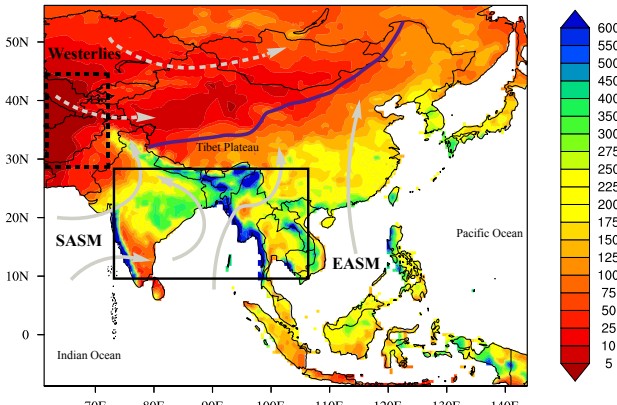

**Figure 1.** Hydrological distribution and climate systems in the Asian monsoon region. The colors indicate the climatological June-July-August mean precipitation (mm month$^{-1}$) in 1981-2010, which are based on the monthly global land-surface precipitation of Global Precipitation Climatology Centre Full Data Reanalysis version 7. The solid and dashed black box indicates the relatively wettest area (RWA) and the relatively driest area (RDA), respectively.

Comparing to the monsoon-dominated subregion, there are much less precipitation in the westerlies-dominated subregion. The southwestern part has the least precipitation (dashed black box, hereafter marked as the relatively driest area (RDA)). Here, because westerly wind brings limited moisture to this region, the transport of air mass from its adjacent areas may play a key role in controlling the moisture conditions (Chen et al., 2008). In the monsoon-dominated subregion, precipitation is largely affected by the evolution of the South and East Asian summer monsoon (Wang et al., 2005). The southern part affected by the

SASM has the most precipitation (solid black box, hereafter marked as the relatively wettest area (RWA)). This large uneven precipitation distribution makes the AMR a susceptible region to perturbations, which has large impact to the local environment and society. Understanding the hydrological variation to perturbations and potential mechanisms are both biophysically and socioeconomically important (Dando, 2005).

Volcanic eruptions are one of the important natural forcing that cool the surface (Robock, 2000; Timmreck, 2012) and cause

strong hydrological perturbations especially in monsoon regions (Iles and Hegerl, 2014; Trenberth and Dai, 2007; Zambri and Robock, 2016; Zhuo et al., 2014; Zhuo et al., 2020). Some studies focused on global impact show a significant decrease of Asian summer monsoon precipitation after volcanic eruptions in both observation (Trenberth and Dai, 2007) and model simulations (Iles and Hegerl, 2014; Zambri and Robock, 2016). A few studies, based on both model simulations (Peng et al., 2010; Man et al., 2014; Man and Zhou, 2014) and hydrological proxy reconstructions (Anchukaitis et al., 2010; Gao and Gao,

2018; Zhuo et al., 2014), focused on Asian summer monsoon response to volcanic eruptions. However, most of them only focused on part of the AMR, except that Zhuo et al. (2020) studied temporal and spatial characteristics of the hydrological impact in subregions of AMR, through comparing proxy reconstruction data and models.

Climate impacts of volcanic eruptions depend on the distribution of volcanic aerosols and the associate radiative forcing structures (Haywood et al., 2013; Toohey et al., 2019; Yang et al., 2019). Haywood et al. (2013) reported the potential inversed





climate effects of the interhemispherically asymmetric volcanic aerosol distributions may have on Sahelian precipitation. Further studies found potential reversed climate impacts of interhemispherically asymmetric volcanic aerosol injection (VAI) in China (Zhuo et al., 2014), tropics (Colose et al., 2016) and monsoon regions (Iles and Hegerl, 2014; Liu et al., 2016; Zuo et al., 2019a; Zhuo et al., 2021). These studies were mostly focused on global or regional mean responses, local hydrological variations are rarely studied.

The mechanisms of the hydrological responses in the AMR were roughly investigated. Precipitation can be reduced from a weakening of the summer monsoon after volcanic eruptions (Dogar and Sato, 2019; Liu et al., 2016; Man and Zhou, 2014; Man et al., 2014; Zhuo et al., 2021; Zuo et al., 2019a). This was generally based on qualitative analysis of the altered land-sea thermal contrast. ITCZ moving toward a warmer hemisphere with less volcanic aerosol loading leads to inversed climate impacts in two hemispheres (Colose et al., 2016; Haywood et al., 2013; Iles and Hegerl, 2014; Zhuo et al., 2021). NH arid

regions get wetter due to an enhanced cross-equator flow and a monsoon-desert coupling mechanism after SHVAI and NHVAI (Zuo et al., 2019b). These cannot fully explain mechanisms of local precipitation responses to volcanic eruptions in subregions of the AMR, as regional responses and local feedback processes were not considered. Zhuo et al. (2021) indicates a dynamical response to VAI and a subsequent physical feedback of local cloud leading to a decreased precipitation in the SASM region. Responses in different subregions of the AMR and related mechanisms need further investigation.

In order to investigate mechanisms of local hydrological responses in subregions of the AMR to different hemispheric VAI, we perform spatio-temporal analyses on multi model ensemble mean of last millennium (LM) experiment from the paleoclimate modelling intercomparison project phase 3 (PMIP3)/coupled model intercomparison project phase 5 (CMIP5). This study aims to answer the following questions: what different hydrological impacts do hemispheric volcanic aerosol injections have in different subregions of AMR? What is the mechanism behind the local hydrological responses to hemispheric VAI?

After this introduction, we describe the data and methods in section 2, followed by our results and discussions in section 3. In section 4, we give our summary and conclusions.

## 2   Data and methods

### 2.1   model data

Nine models participated in the last millennium experiment of PMIP3/CMIP5 (Schmidt et al., 2011). Two different volcanic

forcing index i.e. GRA (Gao et al., 2008) and CEA (Crowley et al., 2008) were freely chosen in the model simulations. Zhuo et al. (2020) calculated two multi-model ensemble means (MMEMs) based on six ensemble members of four models separately adopting the GRA and CEA volcanic forcing index, and compared with proxy reconstruction data. Results indicate the reliability of MMEMs on reproducing the hydrological effects of volcanic eruptions in southern Asian monsoon region. Since similar patterns are shown between two MMEMs, and significant and pronounced patterns are shown in the MMEM

with model members adopting the GRA volcanic forcing (GRA-based MMEM), in this study, we further use it to investigate the mechanism of the hydrological impacts of VAI in subregions of the AMR. The model ensemble members employed in





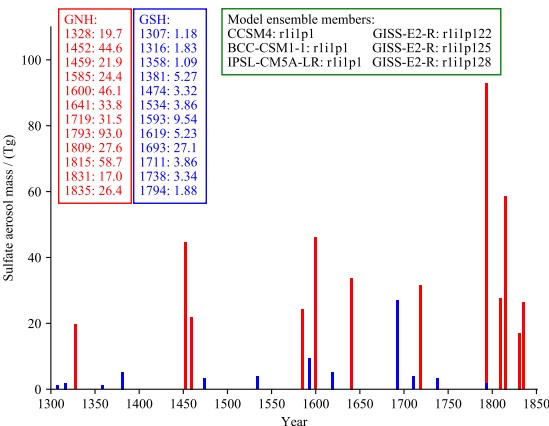

**Figure 2.** Volcanic years and the hemispheric volcanic aerosol injection magnitudes (Tg) in GNH (red lines and texts) and GSH (blue lines and texts) classifications. Model ensemble members used in this chapter are shown in the green box.

the GRA-based MMEM are shown in the green box of figure 2. The same as in Zhuo et al. (2020), we choose the data in 1300-1850 CE and calculate the MMEM after regridding the model outputs to 2.5° × 2.5° spatial resolution.

## 2.2 Volcanic classifications

Following Zhuo et al. (2020), we pick out volcanic events in 1300-1850 CE that have larger northern hemisphere volcanic aerosol injection (NHVAI) than that of 1991 Mount Pinatubo (17 Tg SO2 based on the GRA volcanic forcing index) as GNH classification. To explore reversed hydrological impacts of interhemispherically asymmetric VAI, another classification, with volcanic events in 1300-1850 CE that only have southern hemisphere volcanic aerosol injection (SHVAI), is constructed as GSH classification. Just by coincidence, based on these two criterions, it includes 12 volcanic events in each of the classifi-

cations. Figure 2 shows the years and aerosol magnitudes of the volcanic events. Comparing to the GNH classification, the aerosol magnitudes of volcanic events in the GSH classification are much smaller. This can result in a limited climate impact in the GSH classification, thus might not be sufficient to show reversed hydrological impact of interhemispherically asymmetric VAI. However, since both classifications have 12 volcanic events, the GSH classification is sufficient to serve as a reference classification without NHVAI.

## 2.3 Analysis indices

Following Zhuo et al. (2020), we use a Palmer Drought Severity Index (PDSI, Palmer, 1965) to indicate hydrological conditions. PDSI is calculated from model precipitation and temperature data, together with latitude and water-holding capacity (WEBB et al., 2000), using the MATLAB program produced by Jacobi et al. (2013). It represents normal conditions when





PDSI is between -0.5 and 0.5, and indicates incipient drought when PDSI falls below -0.5 and wet spell when PDSI goes above

95 0.5.

  Previous studies suggest that decreased precipitation in monsoon region is result from a weakened monsoon circulation after volcanic aerosol injections, which based on qualitative analysis of a weakened thermal contrast between the land and the ocean (Man and Zhou, 2014; Man et al., 2014). In this study, we adopt two indices to better quantify the East and South Asian summer monsoon (EASM and SASM) variation. Following the recommendation in Wang et al. (2008), to assess the strength of

EASM, we calculate the East Asian summer monsoon index (EASMI) as the difference of the zonal wind at 850 hPa between over the region 5º - 15º N, 90º - 130º E and 22.5º - 32.5º N, 110º - 140º E (Wang and Fan, 1999), as it outperforms other 24 indices in reflecting the summer precipitation distribution. For the South Asian summer monsoon index (SASMI), we used the definition of Webster and Yang (1992), which is defined as the difference between the zonal wind at 850 hPa over the region 0º–20ºN, 40º-110ºE and the zonal wind at 200 hPa over the region 0º–20ºN, 40º–110ºE. This index is widely used to assess the

large-scale intensity of the southern ASM.

  The moisture transport is reflected by the vertically integrated moisture transport (IVT) and its divergence (IVTD). We calculate the IVT using the following equation:

$$IVT = (1/g) \int_{surface}^{modeltop} qv\,dp \tag{1}$$

where g is the acceleration due to gravity, q is specific humidity, v is the horizontal wind vector, and p is pressure. The vertical

integration of the equation is performed from the surface to the model top.

### 2.4 Superpose epoch analysis with Monte Carlo model test

The Superposed epoch analysis (SEA, Haurwitz and Brier, 1981) method is used to study climate responses to the classified volcanic eruptions. We present 11 years (the eruption year, 5 years before and after the eruption) of the temporal analysis. To test the significance of the results, Monte Carlo model tests (Adams et al., 2003) are performed with 10000 resampling

processes for each year, based on the null hypothesis that there is no relationship between volcanic eruption and climate variation. Significant results at the 95% and 99% confidence levels are identified when SEA results exceed the 95% and 99% range of the Monte Carlo sample. For spatial distribution of the response, since the largest hydrological impacts emerge in the eruption year (year 0) (Zhuo et al., 2020), we present anomalies in the eruption year with respect to the mean of five years before the eruption (Adams et al., 2003; Zhuo et al., 2014). Similar Monte Carlo model tests (Adams et al., 2003) are performed

for significance tests, but with 1000 resampling processes for anomalies of each grid in the eruption year.

### 2.5 Person cross correlation analysis and mechanism exploration

To explore mechanisms of the hydrological effects, we analyze firstly the correlation relationship between temperature, precipitation and the radiation, heat and evaporation related variables, using the widely used Person cross correlation value (r) as the indicator. We calculate r in each grid between variables along the 11 years before and after the aerosol injection, and then





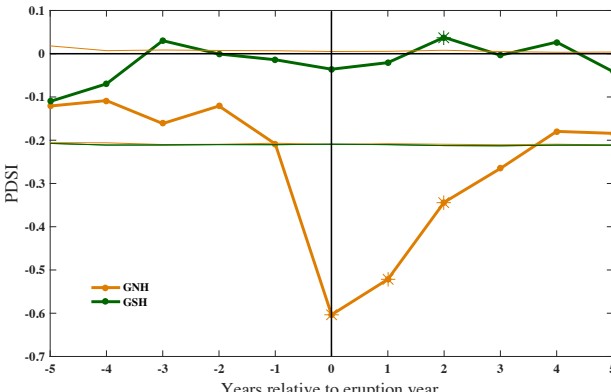

**Figure 3.** Temporal SEA results of summer JJA mean PDSI corresponding to the GNH (yellow line) and GSH (green line) volcanic classifications in 1300-1850 CE in the Asian monsoon region. The thinner lines stand for the relative Monte Carlo model results at the 95% confidence level. The asterisks represent the years that passed the Monte Carlo model tests at the 99% confidence level. Year 0 represents the identified eruption year by volcanic forcing index, negative and positive years represent relative years before and after the eruption.

calculated the average r value of the Asian monsoon region. Hereafter, to explore the mechanism of different hydrological responses in different regions, we show anomalies in the eruption year and compare their spatial patterns of the highly correlated variables.

## 3 Results and discussions

### 3.1 Hydrological responses to NHVAI and SHVAI

Hydrological responses to the classified volcanic eruptions are shown by temporal and spatial SEA results of PDSI. Figure 3 shows the hydrological response to two volcanic classifications in the Asian monsoon region. In the GNH volcanic classification, PDSI indicates significant drying effect in the eruption year (year 0). This drying effect extends to three years after the eruption (year 3). For the GSH classification, PDSI does not show strong changes, but positive PDSI emerges in year 2 and passed the significance test at the 99% confidence level, which indicates a wetting effect. The limited effect might be caused by the limited aerosol magnitude that injected into SH based on the GRA volcanic forcing reconstruction (Gao et al., 2008). However, results in the GSH classification evidently indicate a large difference between with and without volcanic aerosol injection in the northern hemisphere.

Figure 4 further shows the spatial patterns of PDSI in the eruption year when it has the largest drying effect after NHVAI (Fig. 3). In the GNH classification (Fig. 4a), PDSI indicates significant drying effect in a large part of the Asian monsoon region. The strongest drying effect emerges in the southern part of the region (solid black box), while the strongest wetting effect is concentrated in the south-western part of the region (dotted black box). This is exactly opposite to the climatological hydrological conditions in the areas where the RWA and RDA locate. In the GSH classification (Fig. 4b), different from that



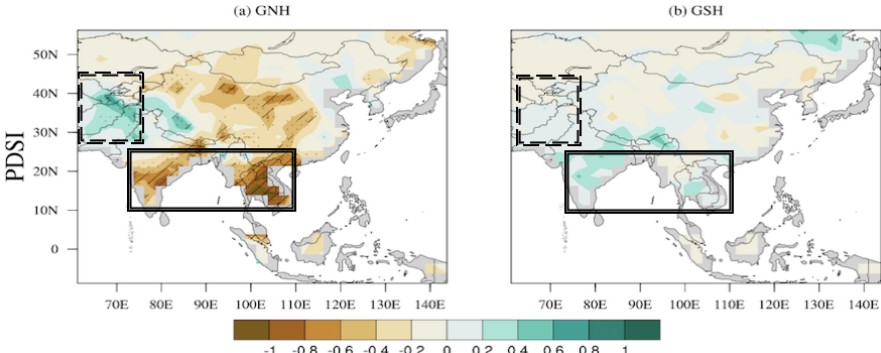

**Figure 4.** Spatial distribution of the Palmer Drought Severity Index (PDSI) anomalies in the eruption year with respect to the mean of five years before the eruption. The solid and dashed black box indicates the relatively wettest area (RWA) and relatively driest area (RDA) as shown in figure 1, respectively. Black dots and slashes indicate significant results at the 95% and 99% confidence level.

in the GNH classification, PDSI shows a wetting effect in the RWA, while a slight wetting effect emerges in the RDA. This indicates inversed hydrological effects between NHVAI and SHVAI in the Asian monsoon region.

The temporal and spatial patterns indicate a distinct "wet gets drier, dry gets wetter" hydrological response to NHVAI, which is opposite to the "wet gets wetter, dry gets drier" precipitation response under global warming and is mainly caused by increasing greenhouse gases (Schurer et al., 2020). This agrees with Zhuo et al. (2021) on a decreased SASM precipitation after NHVAI, and also confirms that NHVAI leads to decreased global monsoon precipitation (Zuo et al., 2019a) and wetter global arid regions (Zuo et al., 2019b Zuo et al., 2019b). However, Zuo et al. (2019b) also found wetter global arid regions

after SHVAI, but our results indicate normal wet condition in the RDA after SHVAI (Fig. 4). This normal condition might be result in limited aerosol magnitude in the GSH classification. The problem that different volcanic classifications have different aerosol magnitude also exist in Zuo et al. (2019b). This brings uncertainty on the conclusion. Even though Zhuo et al. (2021) avoided the problem with the same Pinatubo eruption magnitude in both the NHVAI and SHVAI experiment, the precipitation response is still invisible in the RDA in Zhuo et al. (2021). These disagreements indicate that further studies are needed to

understand the hydrological impact of SHVAI in the arid regions.

### 3.2   Correlation analysis

To identify the key factors that affect the hydrological variation, we show correlations between radiation, heat, moisture related variables and near surface air temperature (T) and precipitation (P) in Table 1, using Pearson cross correlation (r) as the indicator. Since only limited effects are shown in the GSH classification, which is likely due to small magnitude of aerosol

injection, we only conduct the correlation analyses on the GNH classification.

Table 1 shows that T correlates highly with radiation and specific humidity, with r reaching to 0.996 and 0.947 between T and upwelling and downwelling longwave radiation (LW), followed by -0.788 and 0.881 between T and top of the atmosphere (TOA) outgoing shortwave radiation (OSR) and specific humidity. P correlates with evaporation (E) and latent heat flux (LHF,





**Table 1.** Mean Pearson cross correlation (r) values between near surface air temperature (T), precipitation (P) and radiation, heat and moisture related variables over the Asian monsoon region. Numbers in italics and in bold are significant at the 95% and 99% confidence level.

| Variables | Abbreviation | T | P |
|---|---|---|---|
| Top of the atmosphere incident shortwave radiation | TOA ISR | *0.678* | 0.255 |
| Top of the atmosphere outgoing shortwave radiation | TOA OSR | ***-0.788*** | -0.212 |
| Top of the atmosphere outgoing longwave radiation | TOA OLR | **0.706** | 0.121 |
| Surface upwelling shortwave radiation | USR | 0.289 | -0.142 |
| Surface downwelling shortwave radiation | DSR | *0.667* | -0.00171 |
| Surface upwelling longwave radiation | ULR | **0.996** | 0.251 |
| Surface downwelling longwave radiation | DLR | **0.947** | 0.459 |
| Evaporation | E | **0.691** | *0.613* |
| Surface upward latent heat flux | LHF | **0.690** | *0.613* |
| Surface upward sensible heat flux | SHF | 0.321 | -0.300 |
| Near-surface relative humidity | RH | 0.00499 | *0.611* |
| Near-surface specific humidity | / | **0.881** | *0.575* |

both r is equal to 0.613) and closely relates to relative humidity (RH, r is equal to 0.611) and specific humidity (r is equal to 0.575). From these correlations, we know that in order to understand the temperature variation, it is important to investigate shortwave and longwave radiation response to volcanic eruptions. For the precipitation variation, the variations of evaporation, latent heat flux and relative humidity need to be checked. Both, temperature and precipitation, are highly correlated with specific humidity, which indicates that the response of the model follows the Clausius-Clapeyron relation. These correlation analyses can identify the key factors of the hydrological variation but is not sufficient to explain the inversed hydrological responses in the RDA and RWA (Fig. 4).

## 3.3 Mechanisms of the hydrological responses to NHVAI

Considering that limited climate impacts are shown due to limited aerosol magnitude in the GSH classification, we focus on the GNH classification in the following discussions to investigate the mechanisms of the hydrological response to NHVAI.

### 3.3.1 Cooling and subsequent dynamical response to volcanic eruptions

Reflected incoming solar radiation by stratospheric volcanic aerosol leads to significant surface cooling and further affect the hydrological process (Robock, 2002; Timmreck, 2012). Figure 5 shows temperature responses in the two volcanic classifications. The largest temperature decrease over ASM land emerges in year 0 in the GNH classification, and the significant cooling extends to year 2 after the NHVAI. For the GSH classification, insignificant cooling is shown in year 0 to year 2 (Fig. 5a). Figure 5b shows the temperature difference between the land and the ocean in the Asian monsoon region. A significant decrease





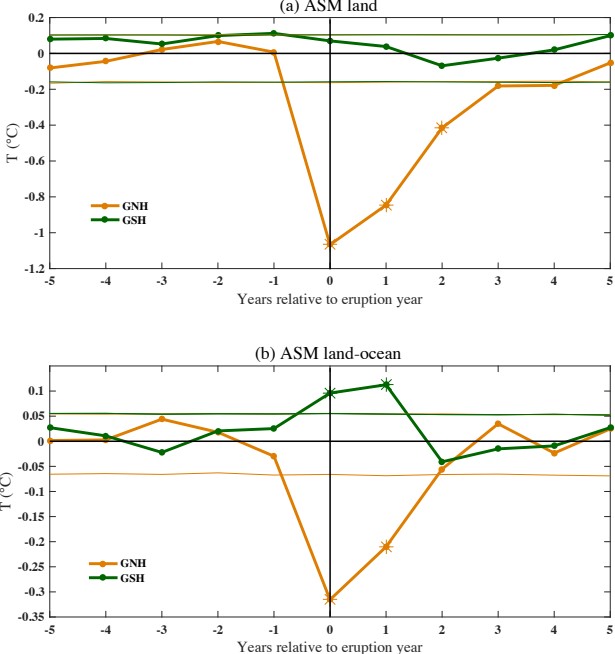

**Figure 5.** Temporal SEA results of summer JJA mean temperature (°C) of land (a) and difference between land and ocean (b) in the Asian monsoon region. The thinner lines stand for the relative Monte Carlo model results at the 95% confidence level. The asterisks represent the years that passed the Monte Carlo model tests at the 99% confidence level. Year 0 represents the identified eruption year by volcanic forcing index, negative and positive years represent relative years before and after the eruption.

in the GNH classification confirms that the NHVAI causes a decreased land-sea thermal contrast in the AMR. In the GSH classification, a significant increase in year 0 and year 1 indicates an increased land-sea thermal contrast after the SHVAI. This quantitative analysis result confirms that volcanic eruption leads to an inhomogeneous cooling between the land and the sea. The opposite responses to the GNH and the GSH classification quantitatively show different impacts of interhemispherically asymmetric VAI, which was also reported in African monsoon region (Haywood et al., 2013) and global monsoon domain (Liu
et al., 2016; Zuo et al., 2019a).

For spatial distributions of the temperature responses, Figure 6 shows that in the GNH classification, the strongest cooling emerges in the RDA, but slight cooling and warming in different parts of the RWA. Oppositely, in the GSH classification, the strongest cooling appears in the RWA. When comparing to the spatial pattern of PDSI (Fig. 6a), in the GNH classification, the area with the strongest wetting effect coincides with the strongest cooling effect in the RDA, while the driest area is identical to
the area with the weakest warming effect in the RWA. This matching relationship between PDSI and temperature also exists in the GSH classification (Fig. 6b). This indicates a strong coupling between temperature variations and hydrological responses to volcanic eruptions.





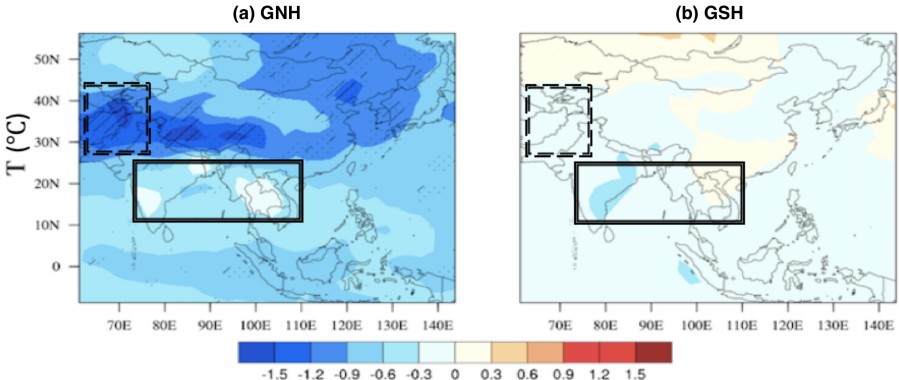

**Figure 6.** Spatial distribution of the temperature anomalies in the eruption year with respect to the mean of five years before the eruption. The solid and dashed black box indicates the relatively wettest area (RWA) and relatively driest area (RDA) as shown in figure 1, respectively. Black dots and slashes indicate significant results at the 95% and 99% confidence level.

Uneven temperature response between the land and the ocean after volcanic eruptions (Fig. 5 and 6) lead to subsequent dynamical response of the climate system. Here, we quantify summer monsoon circulation changes with the EASMI and the

SASMI index. In the GNH classification, the EASMI decreases significantly in year 0, and the significant anomaly lasts to year 3 (Fig. 7a); the SASMI also decreases significantly in year 0 and recovered until year 2 (Fig. 7b). This indicates a significant weakening of the EASM and the SASM. For the GSH classification, the EASMI does not show significant change, while a significant increase of the SASMI in year 0 indicates a strengthening of the SASM. The opposite weakening and strengthening of the SASM after different hemispheric VAI is in agreement with the findings shown in Zhuo et al. (2021).

Changes of the EASM and the SASM show the horizontal motion changes of the atmospheric circulation. The vertical motion changes of the atmospheric circulation are shown by the vertically integrated moisture transport (IVT, vector) and its divergence (IVTD, shaded) in figure 8. Before the volcanic eruptions, the southwest wind transport large amount of moisture from the ocean to the monsoon-dominated subregion, and the RWA is a significant convergence area, while less moisture is transported by the northwest wind to the westerlies-dominated subregion, and the RDA is controlled by the divergence

of moisture flux (Fig. 8a). After the NHVAI, moisture is transported from the ocean and adjacent eastern highlands by the southwest and east wind (Fig. 8b vector) with a strengthened convergence in the RDA (Fig. 8b vector shaded). This results in an enhanced amount and upward transport of moisture, which favors cloud formation and precipitation, and finally results in the significant wetting effect in this area (Fig. 4). In the RWA, the weakened southwest wind deceases moisture transport from the ocean to the land (Fig. 8b vector), and a weakened convergence suppress moisture upward transport (Fig. 8b shaded). This

leads to less cloud formation and precipitation, thus results in the significant drying effect in this area (Fig. 4).





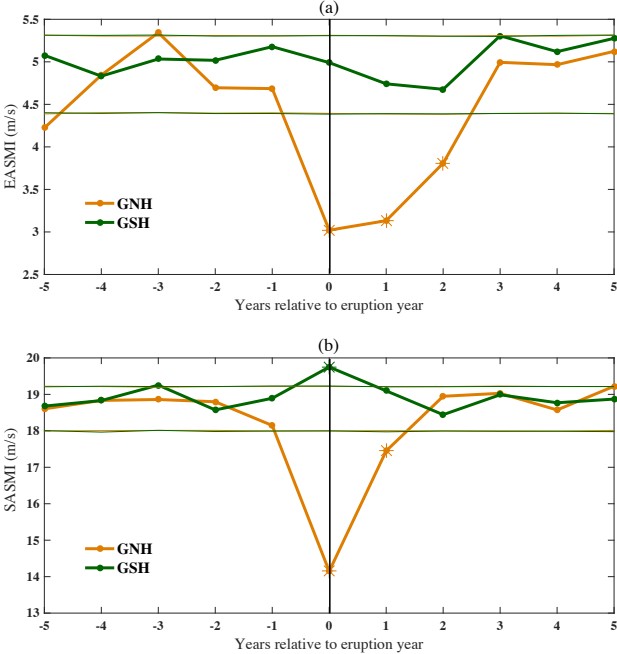

**Figure 7.** Temporal SEA results of the East Asian summer monsoon index (EASMI, m s$^{-1}$, top) and South Asian summer monsoon index (SASMI, m s$^{-1}$, bottom) anomalies. The thinner lines stand for the Monte Carlo model results at the 95% confidence level. The asterisks represent the years that passed the Monte Carlo model tests at the 99% confidence level. Year 0 represents the identified eruption year by GRA volcanic forcing index, negative and positive years represent relative years before and after the eruption.

### 3.3.2 Physical feedbacks of local clouds

Dynamical response of the climate system after VAI leads to changes of cloud cover in different areas, this has corresponding physical feedbacks, which causes different temperature and precipitation variations in different areas. Volcanic sulfate aerosols in the stratosphere reflect solar radiation (SR) at top of the atmosphere (TOA). In clear-sky conditions without taking clouds

into consideration, a significant increase of the TOA OSR indicates an increased reflection of SR after the NHVAI (Fig. 9a). The reflected SR is relatively homogeneous along the same latitude band but decreases from low latitude to high latitude in the northern hemisphere, and more SR is reflected in the RWA comparing to the RDA. This indicates the direct radiative effect of latitude-dependent volcanic aerosols. In comparison, limited variations of OSR emerges in the GSH classification (Fig. 9b), indicating a different pattern without NHVAI. However, the full-sky TOA OSR shows inhomogeneous distribution in different

areas (Fig. 10). Specifically, a stronger reflection of SR emerges in the RDA (Fig. 10a), resulting in a stronger cooling in this area (Fig. 6a). Inconsistent changes of OSR occur in different parts (Fig. 10a), leading to inconsistent temperature changes, with slight cooling or warming in different parts of the RWA (Fig. 6a). The temperature response in different areas reflect the impact of local cloud. As shown in figure 10a, the cloud area fraction (C) increases significantly in the RDA but decreases significantly in the RWA. The TOA OSR is different in the clear-sky condition compared to that in the full-sky condition, and

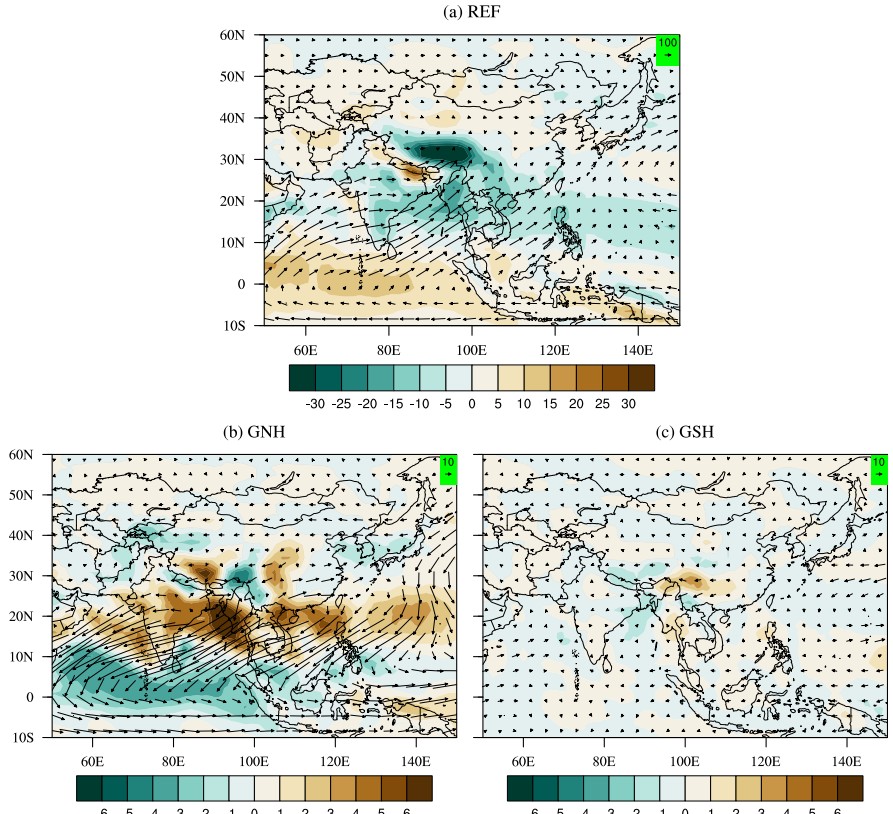

**Figure 8.** JJA mean vertically integrated moisture transport (IVT, vector, kg m$^{-1}$ s$^{-1}$) and its divergence (IVTD, shaded, kg m$^{-2}$ s$^{-1}$) in five years before the eruption (a, REF) and the anomalies in the eruption year of the GNH (b) and GSH (c) classification.

the spatial variation of the full-sky TOA OSR is consistent with the spatial variation of temperature and cloud. These suggest that the regional surface temperature variation is not just due to the direct radiative effect of stratospheric volcanic aerosols, but more dominated by the radiative effect of the subsequently formed atmospheric clouds in different areas.

    The temperature variations further affect the hydrological process. Precipitation is closely related to evaporation (E) and relative humidity (RH) (Table 1). The spatial pattern shows an increase of evaporation in the RDA but a significant decrease

in the RWA. RH increases significantly in the RDA but decreases significantly in the RWA (Fig. 8a). The model follows the Clausius-Clapeyron relation, which connects these responses with the temperature variation. In the RDA, along with the cooling, the saturation humidity is decreased. The significant increase of the relative humidity result from the increase of the actual moisture content in the air, which favors the formation of more clouds and precipitation, and results in the wetting effect here. Oppositely, in the RWA, because of the temperature variation, the saturation humidity varies. The significant decrease of

the relative humidity results from the decrease of the actual moisture content. This reduces the formation of local clouds and precipitation, and results in the drying effect.





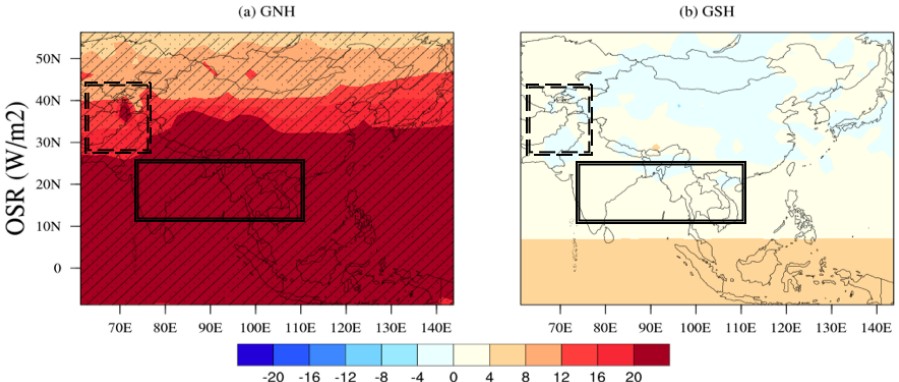

**Figure 9.** Spatial distribution of top of the atmosphere outgoing shortwave radiation (TOA OSR, W m$^{-2}$) anomalies in clear sky in the eruption year with respect to the mean of five years before the eruption. The solid and dashed black box indicates the relatively wettest area (RWA) and the relatively driest area (RDA) as shown in figure 1, respectively. The black dots and slashes indicate the significant results at the 95% and 99% confidence level.

### 3.3.3   Summary of the mechanism and discussion

Based on these results, the mechanism of the hydrological effects of NHVAI in these two representative areas of the monsoon and westerlies-dominated subregions can be summarized as follows: the direct radiative effect of stratospheric volcanic aerosols

affects the atmospheric circulation. The decreased thermal contrast between the land and the ocean results in the weakened EASM and SASM. It changes the moisture transport and the formation of clouds in different areas. The subsequent radiative effect and physical feedback of the local cloud and moisture content lead to different drying and wetting effects in different areas. Specifically, in the RDA, an increased moisture transport from the adjacent south and east areas, together with an enhanced upward motion contribute to the formation of clouds and precipitation, which result in the wetting effect here. In the

RWA, the opposite drying effect results from a decreased moisture transport from the adjacent ocean to the land due to the weakened summer monsoon circulation and weakened upward motion.

Although the mechanism is mainly based on the analysis in the RWA and RDA, where the strongest impact of NHVAI emerges, similar response patterns can be seen in most of the areas, but with a weaker amplitude. They, therefore, reflect the pervasive mechanism of the hydrological response to NHVAI in the monsoon and westerlies-dominated subregions.

The mechanisms summarized here confirms previous studies (Peng et al., 2010; Man et al., 2014; (Iles et al., 2013; Zhuo et al., 2021; Zuo et al., 2019a; Iles and Hegerl, 2014). The decrease of latent heat flux and evaporation over tropical oceans led to the reduction of the summer precipitation in eastern China (Peng et al., 2010). The reduction of monsoon precipitation results in the decreased land-sea thermal contrast and the subsequent weakening of summer monsoon circulation (Iles et al., 2013; Man et al., 2014; Zhuo et al., 2021; Zuo et al., 2019a). (Joseph and Zeng (2011) also found less cooling in areas

near the equator. The regional warming was suggested to be associated with the reduction of clouds, while less evaporation due to the less precipitation further contribute to the regional warming. Zuo et al. (2019b) found a wetting response across

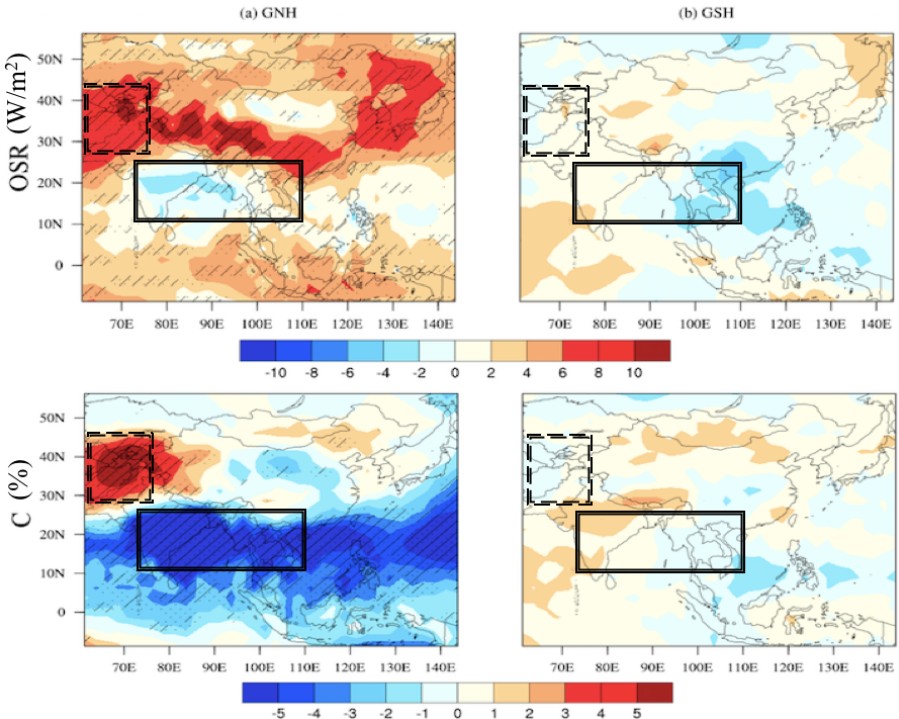

**Figure 10.** Spatial distribution of top of the atmosphere outgoing shortwave radiation (TOA OSR, W m$^{-2}$, top) in full sky and cloud area fraction (C, %, bottom) anomalies in the eruption year with respect to the mean of five years before the eruption. The solid and dashed black box indicates the relatively wettest area (RWA) and relatively driest area (RDA), respectively. The black dots and slashes indicate the significant results at the 95% and 99% confidence level.

arid regions, which is caused by the enhanced cross-equator flow after VAI in the other hemisphere and the monsoon-desert coupling mechanism after VAI in the same hemisphere. This is well reflected by the moisture transport from the adjacent area to the RDA (Fig. 8). Our findings, based on both temporal and spatial analyses, show that the dynamical response changes the

moisture transport and the formation of local clouds, the subsequent radiative effect and physical feedbacks result in different temperature and precipitation responses in different areas. This agrees with Dogar and Sato (2019) on the cloud reduction over the monsoon region, and confirms that both dynamical and physical feedbacks are important to understand regional climate response to volcanic eruptions (Zhuo et al., 2021). As the first study to explore the mechanism of different hydrological responses to volcanic eruptions in the monsoon and westerlies-dominated subregions, we give a comprehensive explanation on

the mechanism of different hydrological response to volcanic eruptions in different areas of monsoon Asia.

### 3.4   Different hydrological responses to SHVAI

Above figures show clear difference between the GNH and the GSH classification. In the GSH classification, oppositely, PDSI indicates significant wetting effect two years after the SHVAI in the Asian monsoon region (Fig. 4), temperature decreases





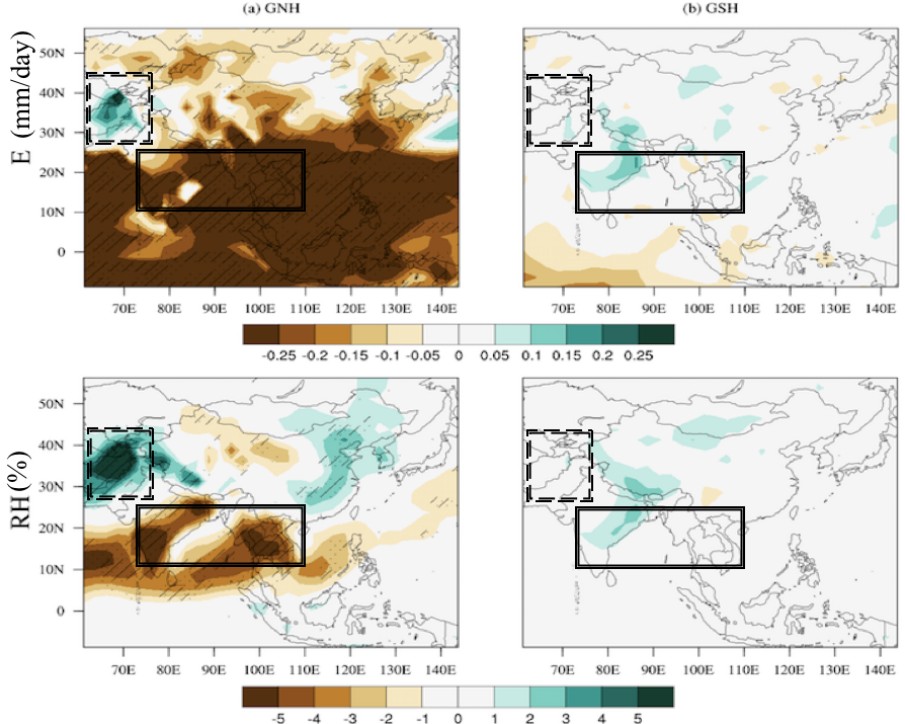

**Figure 11.** Spatial distribution of evaporation (E, mm day$^{-1}$, top) and relative humidity (RH, %, bottom) anomalies in the eruption year with respect to the mean of five years before the eruption. The solid and dashed black box indicates the relatively wettest area (RWA) and relatively driest area (RDA). The black dots and slashes indicate the significant results at the 95% and 99% confidence level.

slightly over the land (Fig. 5a), and the land-sea thermal contrast increases significantly (Fig. 5b), suggesting a cooler effect over the ocean than over the land of the AMR. This contributes to an inversed dynamical response and subsequent physical feedback of the local clouds and precipitation, as shown in the SASM region from most of the spatial patterns (Fig. 4b, 6b and Fig. 8b to 11b). The mechanism of the hydrological response, although inversed, still follows that as summarized in section 3.3.3. This further validates our explanation on the mechanism of the hydrological response to the NHVAI.

The volcanic classifications are based on the volcanic forcing reconstruction (Gao et al., 2008) used in the CMIP5 model simulations (Schmidt et al., 2011), which only identified small aerosol magnitudes for the events in the GSH classification. The large difference of aerosol magnitude between the GNH and the GSH classification brings uncertainty to the conclusion. The small magnitude of volcanic aerosols in the GSH classification has limited climate effect. This makes it imperfect to compare with the significant climate effect shown in the GNH classification. Despite this, results are in good agreement with previous studies. Zhuo et al. (2014), Liu et al. (2016) and Zuo et al., 2019a, based on different criterions of volcanic classifications, pointed out the inversed hydrological effects the asymmetric aerosol loadings may have on monsoon precipitation. Endeavors are also made to understand the mechanism of the hydrological effects over global monsoon regions (Zuo et al., 2019a) and global arid regions (Zuo et al., 2019b). These studies were all based on volcanic classifications that include different magni-





tudes of volcanic aerosols and different numbers of volcanic events. Both can bring large uncertainties to their conclusions.
Zhuo et al. (2021) avoided these uncertainties with same aerosol magnitudes as the 1991 Pinatubo eruption injected into dif-

ferent hemispheres in their sensitivity tests, which also show a reserved hydrological response to NHVAI and SHVAI and is
mostly visible in the SASM region of monsoon Asia. With the same 12 volcanic events coincidentally included in the two clas-
sifications, results in this study provide valuable reference especially on the significant difference between with and without
NHVAI.

## 4   Summary and concluding remarks

We investigate the mechanism of the hydrological responses to volcanic eruptions in different regions of the AMR based on
model output of PMIP3/CMIP5. Hydrological patterns after NHVAI and SHVAI are shown with temporal and spatial analysis
of PDSI. We use correlation analysis to identify key factors that closely relate to climate variation, and compare their spatial
patterns to study the mechanism of the hydrological response to volcanic eruptions in different regions of the AMR.

   After the NHVAI, significant drying effects emerge in the AMR in the eruption year and last to three years after the eruption.
Regionally, the strongest wetting effect emerges in the southwestern part of the AMR while drying effects are concentrated
in the southern Asian summer monsoon region, where the relatively driest area (RDA) and the relatively wettest area (RWA)
locate, forms a "wet gets drier, dry gets wetter" response pattern, that is opposite to the hydrological response to increased
greenhouse gases (Schurer et al., 2020).

   We perform correlation analysis and spatial analyses of related variables to understand the mechanism of the response.
Surface temperature is highly correlated with longwave and shortwave radiation, while precipitation is closely related to evap-
oration, latent heat flux and relative humidity in climate response to volcanic eruptions. Spatial patterns of these variables show
that after NHVAI, temperature gradients decrease between the land and the ocean, this leads to a weakening of the EASM and
SASM that alters the atmosphere circulation. This alters moisture transport and cloud formation process in different regions.
The different regional hydrological responses to volcanic eruptions result from the local physical feedback of atmospheric
clouds, whose distributions are changed due to atmospheric circulation change after aerosol injections. After NHVAI, stronger
cooling emerges over the land than the ocean. This decreases the land-ocean thermal contract, thus weakens the SASM and
EASM. This decreases moisture transport from the ocean to the monsoon-dominated subregions, especially in the RWA, and
together with a suppressed vertical motion, results in the decrease of local cloud formation. Less cloud reflects less SR, which
brings local warming effect. This forms a positive physical feedback with decreased evaporation and relative humidity, thus
lead to the local drying effect. Opposite to the RWA, shifted circulation transport more moisture from adjacent areas to the
westerlies-dominated subregion, and forms more clouds especially over the RDA. This further decrease the shortwave radiation
and bring a significant cooling in the RDA. The cooling and strengthened upward motion by convergence favor condensation.
This forms a positive feedback for more precipitation and causes the local wetting effect.

   Comparing to NHVAI, after SHVAI, it shows a wetting effect and an increased land-sea thermal contrast, and spatially, most
variables show inversed responses especially in the South Asian summer monsoon region. This indicates different hydrological

effects of different hemispheric VAI. However, uncertainties exist with small volcanic aerosol magnitudes in the GSH classification, thus we emphasize on the different hydrological response to with and without NHVAI. Further studies on hydrological responses to SHVAI will contribute to better understanding the different hydrological impact of different hemispheric VAI. Future studies with PMIP4/CMIP6 data (Jungclaus et al., 2017) with updated volcanic forcing reconstruction (Toohey and

Sigl, 2017) can contribute to better understanding on this topic.

Results in this study show opposite hydrological impacts of volcanic eruptions in the driest and wettest areas of AMR, and shed light on mechanisms of the hydrological impact in the westerlies and monsoon-dominated subregions of Asia. Future volcanic eruption might alleviate the uneven hydrological condition that exists between the driest and the wettest area of the AMR. This should be considered in the design of near-term decadal climate prediction and future strategy of local adaptation

to global warming. These results can also give reference to the local hydrological impact of stratospheric aerosol engineering (Simpson et al., 2019) and related mechanisms.

*Code availability.* Post-processing and visualization of data was performed with CDO (https://code.mpimet.mpg.de/projects/cdo) and batch scripts. The scripts are available on request from the corresponding author.

*Data availability.* The PMIP3/CMIP5 data used in this study are from the Deutsche Klimarechenzentrum (DKRZ, https://www.dkrz.de/),
and can be downloaded from the ESGF portal (https://esgf-data.dkrz.de/search/cmip5-dkrz/).

*Author contributions.* ZZ designed the study, analyzed the results and wrote the manuscript. IK and UC supervised and provided support for designing the study. IK provided support for the analysis. All authors contributed to revising the manuscript.

*Competing interests.* The authors declare that they have no conflict of interest.

*Acknowledgements.* This work is supported by China Scholarship Council (CSC). The authors acknowledge the climate modelling groups
listed in figure 2 for producing and making their model output available, the German Climate Computing Center (DKRZ, https://www.dkrz.de/) for making the CMIP5 output and computational resources available.





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
