# Peer review of "Mechanisms of hydrological responses to volcanic eruptions in the Asian monsoon and westerlies-dominated subregions"

_Climate of the Past, 2021_

## Author Comment (AC1)

**Response to Reviewer 1 of manuscript "Mechanisms of hydrological responses to volcanic eruptions in the Asian monsoon and westerlies-dominated subregions" by Zhihong Zhuo et al.**

We are very grateful to Fei Liu for your kind efforts and thoughtful comments, which are very helpful for enhancing the clarity and quality of the manuscript. We have revised the manuscript carefully according to the comments. The list of the reviewer's questions and comments (*in italic*) as well as our responses are listed below. The revised texts are shown in blue.

*Comment on cp-2021-182*
*Fei Liu (Referee)*
*Referee comment on "Mechanisms of hydrological responses to volcanic eruptions in the Asian monsoon and westerlies-dominated subregions" by Zhihong Zhuo et al., Clim. Past Discuss., https://doi.org/10.5194/cp-2021-182-RC1, 2022*
*Review of "**Mechanisms of hydrological responses to volcanic eruptions in the Asian monsoon and westerlies-dominated subregions**" by Zhuo et al*
*Summary and recommendation*
*Understanding the regional hydrological responses to volcanic eruptions at different locations is important to predict the potential climate disasters after future eruptions. This work found a "wet get drier, drier gets wetter" response after the NHVAI, while a significant wetting effect after SHVAI. The relative effects of dynamic and thermodynamics were also investigated. The motivation and results are very interesting, and this manuscript is well organized. I would like to see this work to be published in CP, while before that some Minor revisions are needed.*

Thanks for the positive feedback and comments regarding the study.

*My major concern centers around the discussion on "wet response" or "dry response". This work mainly focused on the PDSI, which response is not only related to precipitation variation but also to temperature change. The increase of precipitation doesn't mean that the PDSI should be increased (Aiguo Dai 2013 Nature Climate Change). In the introduction and main text, the authors should be very carefully to avoid mixing the precipitation and PDSI change.*

Thanks for the comment and the reference. After seeing the difference between the spatial pattern of precipitation and PDSI in Dai (2003), we realized it was inaccurate to conclude on a "wet gets drier, dry gets wetter" response patter based on the PDSI responses, thus we revised the description and discussion a lot regarding this concern. Below shows the revised text on describing the PDSI response to volcanic eruption.

"Figure 3 shows the hydrological response to two volcanic classifications in the Asian monsoon region. In the GNH volcanic classification, PDSI reduces significantly in the eruption year (year 0), and this reduction extends to three years after the eruption (year 3), indicating an intensified aridity after NHVAI. For the GSH classification, PDSI does not show strong changes, but positive PDSI emerges in year 2 and passed the significance test at the 99% confidence level, which indicates a weakened aridity after SHVAI.

Figure 4 further shows the spatial patterns of PDSI in the eruption year when it has the largest drying effect after NHVAI (Fig. 3). In the GNH classification (Fig. 4a), significantly reduced PDSI indicates an intensified aridity in a large part of the Asian monsoon region. The largest reduction of PDSI emerges in the southern part of the region (solid black box), while the largest increase of PDSI is concentrated in the south-western part of the region (dotted black box). This is exactly opposite to the climatological hydrological conditions in the areas where the RWA and RDA locate. In the GSH classification (Fig. 4b), different from that in the GNH classification, PDSI increases in the RWA, while a slight decrease emerges in the RDA.

The PDSI spatial patterns indicate distinct hydrological responses to NHVAI, with a reversed aridity pattern between the RDA and RWA to the climatological conditions. This is

also opposite to the "wet gets wetter, dry gets drier" precipitation response under global warming that is mainly caused by increased anthropogenic greenhouse gases (Schurer et al., 2020)."

The text about discussions on the mechanisms has also been revised, which can be seen in the answer to a following comment regarding line 250.

*Line 14: You mainly focused on the three years after the eruption, which does not belong to the decadal prediction.*

We have removed decadal, use near-term prediction instead.

*Lines 38-40: What are the main results of these works? Are they consistent with your finding?*

We have added some text on the main results of these works, with the revised text "A few studies focused on Asian summer monsoon response to volcanic eruptions, model simulations (Peng et al., 2010; Man et al., 2014; Man and Zhou, 2014) show a reduced precipitation due to a reduced land-sea thermal contrast that in favor of a weakened monsoon circulation, hydrological proxy reconstructions (Anchukaitis et al., 2010; Gao and Gao, 2018; Zhuo et al., 2014) generally agree on the temporal drying trend in the monsoon region, but discrepancies exist in spatial responses to volcanic classifications among different reconstruction data.". Similar to what written here, our findings show consistency on the temporal responses to volcanic eruption, but special responses are more complex, comparisons between model and proxy reconstruction data on these temporal and spatial similarities and discrepancies were investigated and discussed in our previously published paper (Zhuo et al., 2020).

*Line 58: More details of this local cloud feedback are appreciated. Do you mean that the longwave radiation of the cloud will increase the convection?*

We have added more details of this local cloud feedback with following revised text: "Spatial analyses were conducted in Zhuo et al. (2021) in order to understand the mechanism of precipitation responses to volcanic eruptions in the SASM region. Results indicate a dynamical response to VAI, with changed interhemispheric thermal contrast and land-sea thermal contrast, local cloud cover changes in different areas, this leads to subsequent physical feedback on local temperature response, together with the adjusted horizontal and vertical motion of local water vapor, leading to a decreased precipitation in the SASM region after NHVAI.". Zhuo et al. (2021) didn't focus on the longwave radiation of the clouds, instead indicated the physical feedback on shortwave radiation of the cloud. As different local cloud cover changes the local surface shortwave radiation, this contributes to different temperature responses in different areas, and adjust the local horizontal and vertical motion of water vapor and thus changes local precipitation.

*Line 70: The dataset of Ammann et al. 2007 was used in IPSL model. Please check whether you used this model or not?*

We used this model in our analysis, but it won't affect the results, as the Ammann et al. 2007 dataset used the same loading as Gao et al. 2008. This information is written in the volcanic forcing dataset used in the IPSL model simulation. The dataset is from Myriam Khodri who conducted the IPSL model simulation, after I contacted with Jean-Louis Dufresne, who is the corresponding author of the model reference paper. We would like to also express my appreciation to Jean-Louis Dufresne and Myriam Khodri here for their help and generous sharing of the forcing dataset.

*Line 124: I don't know how the correlations are calculated. Did you calculate it among different eruptions or among the 11 selected years? More details are needed.*

It's calculated among the 11 selected years. Maybe we did not write it clearly, we revised the text to "We calculate r in each grid between variables among the selected 11 years before and after the aerosol injection, and then calculated the average r value of the Asian monsoon region.".

*Figure caption 3: Definition of the Asian monsoon region is necessary.*

As suggested, the text has been revised to "…in the Asian monsoon region (8.75°S–56.25°N, 61.25°CE–143.75°E)" in the figure caption.

*Lines 138-139: The reconstructed PDSI response of Asian monsoon to different eruptions was first discussed by Liu et al. 2016 SR. Comparison with this reconstruction analysis is necessary.*

In the revised manuscript, we have added following discussion on the comparison of the PDSI response with Liu et al. (2016) paper: "The reduction of PDSI in the GNH classification agrees on a weakened Asian monsoon with Liu et al. (2016), which showed significant reduction of PDSI in the first year after tropical eruptions and the second year after NH volcanic eruptions. Due to limited aerosol magnitude in the GSH classification, slight increase of PDSI emerges after SHVAI and is only significant in year 2. This also agrees well with Liu et al. (2016), which showed an increased PDSI in the first year from SH volcanic eruptions, although without passing the significance tests.".

*Figure 5: Definitions of these ASM land and ocean regions are needed.*

As suggested, the text has been revised to "…in the Asian monsoon region (land and ocean part in 8.75°S–56.25°N, 61.25°CE–143.75°E)." in the figure caption.

*Line 188: Figure 6 exhibits the temperature anomalies, not the PDSI.*

The typo has been revised to figure 4.

*Fig. 8: Significant test is needed in Figs. 8b and 8c.*

The significance tests of IVT have been conducted and the test results have been added to the revised fig. 8.

*Line 250: I don't think the mechanisms are totally the same. The change of PDSI include both precipitation and temperature related evaporation variations. Previous works mainly focus on the precipitation change.*

We agree that the mechanisms are not totally the same, we thought it confirms these previous works that reflect part of the mechanisms shown in this study. Precious works mostly show that precipitation change can be explained only by the dynamical response, but mechanisms of the hydrological response relates to both precipitation and temperature, thus is related to both dynamical response and local physical feedbacks. In order to make the difference clearer, the discussion text has been revised to: "Previous studies explored the mechanisms of precipitation responses to volcanic eruptions (Peng et al., 2010; Man et al., 2014; Iles et al., 2013; Zhuo et al., 2021; Zuo et al., 2019a; 2019b). The reduction of monsoon precipitation results in the decreased land-sea thermal contrast and the subsequent weakening of summer monsoon circulation (Iles et al., 2013; Man et al., 2014; Zhuo et al., 2021; Zuo et al., 2019a). Our quantitative analysis confirms this on the dynamical response of the climate system to volcanic eruptions. The decrease of latent heat flux and evaporation over tropical oceans led to the reduction of the summer precipitation in eastern China (Peng et al., 2010). Zuo et al. (2019b) found a wetting response across arid regions, which is caused by the

enhanced cross-equator flow after VAI in the other hemisphere and the monsoon-desert coupling mechanism after VAI in the same hemisphere. This is well reflected by the moisture transport from the adjacent area to the RDA (Fig. 8). Joseph and Zeng (2011) found less cooling in areas near the equator. The regional warming was suggested to be associated with the reduction of clouds, while less evaporation due to the less precipitation further contribute to the regional warming. This indicates that regional temperature and precipitation responses relate to changes of local clouds. Our findings, based on both temporal and spatial analyses, show the importance of both the dynamical response and the physical feedback on understanding the mechanisms of hydrological responses to NHVAI. The dynamical response changes the moisture transport and the formation of local clouds, the subsequent radiative effect and physical feedbacks result in different temperature and precipitation responses in different areas."

*Line 295: The RDA region is actually located at central Asia.*
      The region is mostly located at central Asia, but the RDA region does not cover the whole part of central Asia, thus in order to not mix the definition, we used the self-defined RDA in the manuscript.

---

## Author Comment (AC2)

**Response to Reviewer 2 of manuscript "Mechanisms of hydrological responses to volcanic eruptions in the Asian monsoon and westerlies-dominated subregions" by Zhihong Zhuo et al.**

We are very grateful to the anonymous reviewer for your kind efforts and thoughtful comments, which are very helpful for enhancing the clarity and quality of the manuscript. We have revised the manuscript carefully according to the comments. The list of the reviewer's questions and comments (*in italic*) as well as our responses are listed below. The revised texts are shown in blue.

*Comment on cp-2021-182*
*Anonymous Referee #2*
*Referee comment on "Mechanisms of hydrological responses to volcanic eruptions in the Asian monsoon and westerlies-dominated subregions" by Zhihong Zhuo et al., Clim. Past Discuss., https://doi.org/10.5194/cp-2021-182-RC2, 2022*

*This paper uses the PMIP3/CMIP5 past1000 ensemble to investigate how explosive volcanic eruptions affect surface climate in the Asian monsoon region. The paper provides a nice analysis of the hydrological response in different regions within this larger area, and also contrasts the response to predominantly Northern hemisphere eruptions and Southern hemisphere eruptions. The analysis is interesting and clearly described and as such is publishable in this journal. Before this occurs though I have two more major concerns which I would like to see addressed, in addition to some more minor comments.*

Thanks for the positive feedback and comments regarding the study. We have considered the comments carefully and revised the manuscript carefully accordingly. The detailed answers can be seen under the specific comments.

*As also mentioned by reviewer 1, care needs to be taken when talking about these results in the context of the "wet-gets-wetter, dry-gets-drier" paradigm. Schurer et al 2020 (in addition to a number of previous studies) analysed precipitation across the whole tropic and found a detectable response in the wettest and driest regions. I do not think that it is definitely the case that this will also apply when restricting the analysis to only the summer climate of the Asian monsoon region, and particularly not to PDSI over this region. Also the fact that you are analysing PDSI should be taken into account when discussing the link to temperature.*

Thanks for the comment and the reference. After considering both of your comments, we realized it was inaccurate to conclude on a "wet gets drier, dry gets wetter" response patter based on the PDSI responses, thus we revised the description and discussion a lot regarding this problem. Below shows some revised text as revision examples.

"Figure 3 shows the hydrological response to two volcanic classifications in the Asian monsoon region. In the GNH volcanic classification, PDSI reduces significantly in the eruption year (year 0), and this reduction extends to three years after the eruption (year 3), indicating an intensified aridity after NHVAI. For the GSH classification, PDSI does not show strong changes, but positive PDSI emerges in year 2 and passed the significance test at the 99% confidence level, which indicates a weakened aridity after SHVAI.

Figure 4 further shows the spatial patterns of PDSI in the eruption year when it has the largest drying effect after NHVAI (Fig. 3). In the GNH classification (Fig. 4a), significantly reduced PDSI indicates an intensified aridity in a large part of the Asian monsoon region. The largest reduction of PDSI emerges in the southern part of the region (solid black box), while the largest increase of PDSI is concentrated in the south-western part of the region (dotted black box). This is exactly opposite to the climatological hydrological conditions in the areas

where the RWA and RDA locate. In the GSH classification (Fig. 4b), different from that in the GNH classification, PDSI increases in the RWA, while a slight decrease emerges in the RDA.

The PDSI spatial patterns indicate distinct hydrological responses to NHVAI, with a reversed aridity pattern between the RDA and RWA to the climatological conditions. This may counteract the "wet gets wetter, dry gets drier" precipitation response to global warming that is mainly caused by increased anthropogenic greenhouse gases (Schurer et al., 2020)."

The mechanisms relate to both dynamical response and physical feedback, which indicates the link of PDSI to both precipitation and temperature. In order to make the difference clearer, the text about the discussion on mechanism has been revised to: "Previous studies explored the mechanisms of precipitation responses to volcanic eruptions (Peng et al., 2010; Man et al., 2014; Iles et al., 2013; Zhuo et al., 2021; Zuo et al., 2019a; 2019b). The reduction of monsoon precipitation results in the decreased land-sea thermal contrast and the subsequent weakening of summer monsoon circulation (Iles et al., 2013; Man et al., 2014; Zhuo et al., 2021; Zuo et al., 2019a). Our quantitative analysis confirms this on the dynamical response of the climate system to volcanic eruptions. The decrease of latent heat flux and evaporation over tropical oceans led to the reduction of the summer precipitation in eastern China (Peng et al., 2010). Zuo et al. (2019b) found a wetting response across arid regions, which is caused by the enhanced cross-equator flow after VAI in the other hemisphere and the monsoon-desert coupling mechanism after VAI in the same hemisphere. This is well reflected by the moisture transport from the adjacent area to the RDA (Fig. 8). Joseph and Zeng (2011) found less cooling in areas near the equator. The regional warming was suggested to be associated with the reduction of clouds, while less evaporation due to the less precipitation further contribute to the regional warming. This indicates that regional temperature and precipitation responses relate to changes of local clouds. Our findings, based on both temporal and spatial analyses, show the importance of both the dynamical response and the physical feedback on understanding the mechanisms of hydrological responses to NHVAI. The dynamical response changes the moisture transport and the formation of local clouds, the subsequent radiative effect and physical feedbacks result in different temperature and precipitation responses in different areas."

*As correctly acknowledged by the authors, there have already been a number of other studies analysing the response to the monsoon regions to large volcanic eruption. Although many have been cited here (e.g. lines 38-41, 51-55) I think that the paper would really benefit with a more detailed description of what some of these key papers found, in particular highlighting what exactly is novel here.*

Thanks for the comment, we added more detailed description on these studies. As suggested, more detailed descriptions have been added in order to better highlighting the novelty of our study.

Close to lines 38-41, the text has been revised to "A few studies focused on Asian summer monsoon response to volcanic eruptions, model simulations (Peng et al., 2010; Man et al., 2014; Man and Zhou, 2014) show a reduced precipitation due to a reduced land-sea thermal contrast that in favor of a weakened monsoon circulation, hydrological proxy reconstructions (Anchukaitis et al., 2010; Gao and Gao, 2018; Zhuo et al., 2014) generally agree on the temporal drying trend in the monsoon region, but discrepancies exist in spatial responses to volcanic classifications among different reconstruction data.".

Close to lines 51-55, the text has been revised to "The mechanisms of the hydrological responses in the AMR were roughly investigated. Precipitation can be reduced from a weakening of the summer monsoon after volcanic eruptions (Dogar and Sato, 2019; Liu et al., 2016; Man and Zhou, 2014; Man et al., 2014; Zhuo et al., 2021; Zuo et al., 2019a). This was

generally based on qualitative analysis of the altered land-sea thermal contrast. ITCZ moving toward a warmer hemisphere with less volcanic aerosol loading leads to inversed climate impacts in two hemispheres (Colose et al., 2016; Haywood et al., 2013; Iles and Hegerl, 2014). These studies focused on mechanisms of instant precipitation response, which does not reflect the degree of dryness after volcanic eruptions. And the analysis was conducted holistically over the investigated region. Zuo et al. (2019b) adopted both precipitation and drought reconstruction data in their analysis, all of them showed wetter conditions in arid regions after all types of volcanic eruptions, which is due to an enhanced cross-equator flow and a monsoon-desert coupling mechanism after SHVAI and NHVAI. However, moisture budget analyses were also conducted holistically over the hemispheric arid regions in Zuo et al. (2019b). These cannot fully explain mechanisms of local hydrological responses to volcanic eruptions, as regional responses and local feedback processes were not considered. Spatial analyses were conducted in Zhuo et al. (2021) in order to understand the mechanism of precipitation responses to volcanic eruptions in the SASM region. Results indicates a dynamical response to VAI, with changed interhemispheric thermal contrast and land-sea thermal contrast, local cloud cover changes in different areas, this leads to subsequent physical feedback on local temperature response, together with the adjusted horizontal and vertical motion of local water vapor, leading to a decreased precipitation in the SASM region after NHVAI. No spatial analysis is conducted in order to understand the mechanisms of hydrological responses to volcanic eruptions in areas of the AMR in different monsoon and westerlies-dominated subregions.

This study tries to fill in the gap to investigate mechanisms of local hydrological responses in monsoon and westerlies-dominated subregions of the AMR to different hemispheric VAI. We perform spatio-temporal analyses on multi model ensemble mean of last millennium (LM) experiment from the paleoclimate modelling intercomparison project phase 3 (PMIP3)/coupled model intercomparison project phase 5 (CMIP5).".

*Minor comments:*
*In the abstract (and throughout) please ensure all acronyms are defined (e.g. RWA, RDA, EASM, SASM).*
      Accordingly, for RWA and RDA, the abstract has been revised to "drying effect in the relatively wettest area (RWA) but significant wetting effect in the relatively driest area (RDA) of the AMR.". For EASM and SASM, since it's not used again, we have replaced them with the full text as "…weakened East Asian summer monsoon and South Asian summer monsoon".

*L13 – To avoid misunderstanding - I think it would help to clarify that effects of future volcanic eruptions will only be a temporary, e.g. "future volcanic eruptions may temporary alleviate..."*
      The text has been revised as suggested.

*L17 – would it be possible to include in figure 1 – what the boundaries are for your definition of the, EASM, SASM. Although not strictly necessary, I think this could help many readers understand the results more quickly.*
      The boundaries of EASM and SASM can be different according to different definitions from previous studies, thus it's hard to draw the exact boundaries in the figure, but we wrote in the text that according to Chiang et al. (2017), in the monsoon-dominated region, the EASM and the SASM is usually separated by 100°E longitude.

*L20 – how is the modern Asian summer monsoon limit defined (or alternatively give a citation where it is defined)*

It's referenced from Chen et al. (2018) as cited at the end of this sentence, but based on this comment, the reference citation has been added right after here as "…the modern Asian summer monsoon limit (Chen et al., 2018)".

*Section 2.1 – I think that the model selection section could be better explained. Was this entirely based on the work of Zhou et al 2020? If so this should be made clearer. The GRA forcing in GISS was implemented approximately twice as strong as it should have been, see e.g. errata and comments here: https://data.giss.nasa.gov/modelE/cmip5/ Could this be why the GRA MMM is more significant?*

Yes, as understood, the model selection is entirely based on the work of Zhuo et al. (2020), in which the model data were firstly evaluated in comparison to proxy data. We have made it clearer with revised text as "The green box of figure 2 shows model ensemble members employed in the GRA-based MMEM, which are the same as in Zhuo et al. (2020)." This GISS double forcing problem was already discussed in Zhuo et al. (2020). The GRA MMM is more significant partly because of it, but also because GRA forcing is larger than CEA forcing, as in the reconstruction of CEA forcing, 2/3 power scaling was applied to all values greater than or equal to 0.200 AOD, considering the collisions between aerosol particles result in size increases, shortwave radiative forcing is reduced by the 2/3 power (Crowley et al., 2013), the smaller CEA forcing itself also contribute to less significant results when comparing to GRA MMM results.

*Figure 2 – this should make it clear in the caption that this is just for the GRA dataset.*

As suggested, the text has been revised to "classifications based on the GRA volcanic forcing." in the figure caption.

*Section 2.2 – how were the SHVAI eruptions classified – was there a threshold? And are the NHVAI only defined based on a NH threshold? Does this necessarily mean that the NHVAI is larger than the SHVAI? More details and justification are needed for this section.*

There is no threshold for the SHVAI classification, the definition is that all those eruptions that only have southern hemisphere volcanic aerosol injection were selected, as the number of events is limited and especially the magnitude of these events is quite small according to the GRA volcanic forcing, thus it is hard to apply the threshold as that in the NHVAI classification. Yes, the NHVAI is larger than SHVAI. As we wrote in the beginning that the classification is following Zhuo et al. (2020). We also noted in line 85-88 in the end of this paragraph on the potential limited climate impact of this much smaller aerosol magnitude, that's why we also noted in line 88-89 that the GSH classification is serve as a reference classification without NHVAI, and we mainly focused on the mechanisms of hydrological responses to NHVAI in section 3.3, and used a separate part (section 3.4) to fucus on the SHVAI, and their difference to the NHVAI.

*Section 2.6 – did you mean Pearson correlation?*

Yes, the typo has been revised.

*Figure 3- Make it clear in the caption which region this refers to.*

As suggested, the text has been revised to "…in the Asian monsoon region (8.75°S–56.25°N, 61.25°CE–143.75°E)." in the figure caption.

*Can you explain why the SH eruptions seem to be significantly wet even before the eruption (e.g. year -3)? Given that the PDSI before the eruption seems so different between the GSH*

*and GNH can you really be confident the value for the GSH in year +2 is significant, and due to the eruption?*

Year -3 passed the significance test at the 95% confidence level, year 4 also passed at the 95% significance level, but year 2 passed at the 99% confidence level, so we only noted that year 2 is significantly wet, and we discussed, in line 150-152, the uncertainty that due to the limited magnitude of the classification shown in this study and also from previous studies lie Zuo et al. (2019b), but we also added discussed that this can be the case, as previous studies also showed the wetting effect after SH volcanic eruption, especially in Zhuo et al. (2021) study, which used the same Pinatubo strength for simulations, and their results reversed drying and wetting effect after the NH and SH eruption.

*Figure 4 – can you describe what significance test you performed here? Is it also possible in this and subsequent figures to make the stippling clearer?*

The significance test is based on Monte Carlo model test, the details are described in section 2.4, which is a method following Adams et al (2003) and has also been used in Zhuo et al. (2014, 2021). We added this method in the figure caption and revised the text to "Black slashes and cross signs indicate significant results that passed the Monte Carlo model tests at the 95% and 99% confidence level". In order to make it clearer, the figures have been replotted with pdf format, and also the stippling format has been changed to slashes and cross signs instead of dots and slashes.

*Section 3.3 – says that the results will only discuss the NHVAI – yet go onto to also discuss the SHVAI.*

At the beginning of section 3.3, it says that we focus on the GNH classification, not only discuss the NHVAI. We also discussed the SHVAI, in order to show the large difference clearly between GNH and GSH, this can contribute to better understand the significant impact of NHVAI and its potential mechanism. The section tittle reflects the main content well, as most discussions focus on the NHVAI, and section 3.3.3, the last part of this section summarized the mechanisms of the hydrological responses to NHVAI.

*Figure 8 – What is the scale for the arrows? Is it the same in all figures? Also the caption should make it clear that the color scales are different.*

The scale for the arrows shows in the green box at top-right corner of the figure, it's 100 in (a) REF at the top panel, but 10 in (b) GNH and (c) GSH at the bottom panel. We added in the caption that "The scale for the arrow shows in the green box at top-right corner of the subfigure. Note the scales of the colors and arrows are different between the top and bottom panel of the figure.

*Figure 9 – why is the feature in figure b such a clear line – is this expected? In panel b why is there no effect at all in the NH (whereas there is in the SH in panel a) – is this expected given the definition of SHVAI?*

This is the outgoing shortwave radiation in the clear sky, which reflects the direct radiative effect of volcanic aerosols, thus it's reasonable that it's quite uniform along the longitude, as the aerosols are considered to be quickly distributed across the globe and uniformly distributed along the longitude. We have doubled checked, the clear line is due to the scale of the color bar, they are all within the range of 4 to 8 W/m$^2$ at different longitudes, thus becomes a line in the figure. The limited effect in the NH in Panel b is due to the definition of SHVAI, as no aerosol injected into the NH based on GRA forcing, thus there's no direct radiative effect from volcanic aerosols in the NH.

*Line 268 – should this refer to figure 3?*
Yes, we revised it to figure 3.

---

## Editor Decision (ED1)

**Response to editor's comments**
**"Mechanisms of hydrological responses to volcanic eruptions in the Asian monsoon and westerlies-dominated subregions" by Zhihong Zhuo et al.**

Upon re-reading this manuscript and puzzling over what seems like odd figures, I realize that Figure 2 is really rather inconsistent with Colose et al 2016 Figure 1.

[Figure]

This becomes important for this whole paper because in the present Zhou analysis, there is really only 1 example of a 'large' southern hemisphere eruption. I don't really understand why this is, so I replotted the volcanic forcings for GISS (these are in AOD space, not Tg loading space, but the former is scaled to apply forcing to climate models).

[Figure]

If we zoom in on the 1450's, we see NH (20-90N), TR (20S-20N), and SH(90S-20S) look like this… so why is it classified as GNH?

[Figure]

For my previous comments…

*It is difficult to get into the details of tropical precipitation without speaking of ENSO – Zanchettin et al 2022 (doi: 10.5194/gmd-15-2265-2022) goes through VolMIP (the updated versions of the PMIP3/CMIP5 models specifically checking out volcanic forcing), and should be referenced. Khodri 2017 needs to be cited and integrated too (doi:10.1038/s41467-017- 00755-6). Volcano-ENSO links which will have hydroclimate impacts as well. It is a bit odd quite frankly not to reference VolMIP at all. At least a paragraph should be added.*

This is a follow-up study after Zhuo et al., 2020, in which we analyzed the climate response to volcanic eruptions with detailed proxy-model comparison, and we used one paragraph discussing the volcano-ENSO links, in which we noted that "Following the method in Iles et al. (2013), we test this uncertainty by repeating the SEA analysis after regressing out the effect of ENSO. Consistent with Iles et al. (2013) and Iles and Hegerl (2014), it only results in a lower amplitude response, but the temporal and spatial patterns remain largely unchanged." (Zhuo et al., 2020). This study is focused on explaining the mechanism of the response patterns identified in Zhuo et al. (2020), in order to avoid redundancy, we didn't discuss ENSO again in this study, but we added some text in the final part to emphasis the necessity of future studies on it. We agree that, like PMIP4/CMIP6, VolMIP should also be mentioned. Considering these comments, we added following concluding remarks as shown in line 334-345 in the revised manuscript:

Except for forcing inputs, uncertainties of the hydrological responses can also come from internal variability, especially, as discussed in Zhuo et al. (2020), the initial state of the El Niño-Southern Oscillation (ENSO) and its response to volcanic eruption. Studies tend to consensus on a El Niño tendency after tropical and NH volcanic eruptions (Khodri et al., 2017; Liu et al., 2022, Stevenson et al., 2016), but this can be an overestimation of the forced response relative to natural ENSO variability (Dee et al., 2020). More studies are also needed to understand the ENSO response to SH volcanic eruptions. Besides, the interaction between post-eruption ENSO and monsoon precipitation varies in different monsoon regions. A weakened African monsoon due to post-eruption cooling in Africa leads to the El Nino response after tropical eruptions (Khodri et al., 2017), but a more frequent occurrence of El Niño in the first boreal winter after eruptions lead to an enhanced EASM (Liu et al., 2022). The interaction among ENSO, monsoon and volcanic eruptions remains unclear. The Model Intercomparison Project on the climatic response to Volcanic forcing (VolMIP, Zanchettin et al., 2016) and its potential future phases with improved protocol addressing the pre-eruption ENSO state (Zanchettin et al., 2020) can be valuable resource to investigate these questions.

Above, the authors do not mention the VolMIP results that specifically explore the impact of the initial ENSO state on the simulated response. https://gmd.copernicus.org/articles/15/2265/2022/ It is too important and relevant to the discussion. I disagree that the relatively small ensemble used here (and in the previous paper) is an adequate stand-in for ENSO discussion. Its needs its own section plus context woven throughout the text. The models used here DO INDEED appear in Davide's analysis… except that there was intentional sampling across a variety of initial conditions. It should be pretty straight-forward to 'look up' the range of responses, the implications for precip, convergence, temperature, etc. and carry on. This paper really does have

insights into how initial condition projects onto temperature, heat fluxes, etc. And it is a MUCH larger sampling of volcanic forcing which is rather ephemeral, and sometimes hard to tease out beyond the background 'noise'/variability.

*The change quoted below makes this paragraph make less sense. A precipitation epoch analysis is exactly a 'degree of dryness' analysis. But that turn of phrase really doesn't make very much sense. Also, not all arid places get wetter after volcanism. Take Manning et al 2017 (https://doi.org/10.1038/s41467-017-00957-y) – volcanoes made Nile Valley drier.*

*"ITCZ moving toward a warmer hemisphere with less volcanic aerosol loading leads to inversed climate impacts in two hemispheres (Colose et al., 2016; Haywood et al., 2013; Iles and Hegerl, 2014; Zhuo et al., 2021). NH arid regions get wetter These studies focused on mechanisms of instant precipitation response, which does not reflect the degree of dryness after volcanic eruptions. And the analysis was conducted holistically over the investigated region. Zuo et al. (2019b) adopted both precipitation and drought reconstruction data in their analysis, all of them showed wetter conditions in arid regions after all types of volcanic eruptions, which is due to an enhanced cross-equator flow after SHVAI and a monsoon- desert coupling mechanism after SHVAI and NHVAI(Zuo et al., 2019b). NHVAI. However, moisture budget analyses were also conducted holistically over the hemispheric arid regions in Zuo et al. (2019b). These cannot fully explain mechanisms of local precipitation hydrological responses to volcanic eruptions in subregions of the AMR, as regional responses and local feedback processes were not considered. Zhuo et al. (2021) indicates Spatial analyses were conducted in Zhuo et al. (2021) in order to understand the mechanism of precipitation responses to volcanic eruptions in the SASM region. Results indicate a dynamical response to VAIand a , with changed interhemispheric thermal contrast and land- sea thermal contrast, local cloud cover changes in different areas, this leads to subsequent physical feedback of local cloud on local temperature response, together with the adjusted horizontal and vertical motion of local water vapor, leading to a decreased precipitation in the SASM region after NHVAI. No spatial analysis is conducted in order to understand the mechanisms of hydrological responses to volcanic eruptions in areas of the AMR. Responses in different subregions of the AMR and related mechanisms need further investigation."*

We further refined this part as follows:

ITCZ moving toward a warmer hemisphere with less volcanic aerosol loading leads to inversed climate impacts in two hemispheres (Colose et al., 2016; Haywood et al., 2013; Iles and Hegerl, 2014; Zhuo et al., 2021). With moisture budget analyses over the hemispheric arid regions, Zuo et al. (2019b) showed wetter conditions in NH arid regions due to an enhanced cross-equator flow after SHVAI and a monsoon-desert coupling mechanism after NHVAI. However, these analyses cannot fully explain mechanisms of local hydrological responses to volcanic eruptions, as regional responses and local feedback processes were not considered. Based on spatial analysis, Zhuo et al. (2021) showed that dynamical responses to NHVAI change local cloud cover. A subsequent physical feedback of local temperature and adjusted horizontal and vertical

motion of local water vapor lead to a decreased precipitation in the SASM region. Responses in different subregions of the AMR and related mechanisms need further investigation.

Here is a place where you could add an extra sentence wrt ENSO.

We agree that not all arid places get wetter after volcanism. Zuo et al. (2019b) took NH arid regions as a whole, which ignored the potential difference in different local areas. This is why it is important to investigate into local hydrological responses. We always point out that the response is in the areas of the AMR, like that written in the abstract "with an intensified aridity in the relatively wettest area (RWA) but a weakened aridity in the relatively driest area (RDA) of the AMR".

*Figure 4,6,8 – I have a strong preference for hatching to be on the insignificant data because it makes the significant data easier to see instead of obscured. I think this is the opposite way of this figure. I suggest doing this the opposite way.*

We tried to replot the figures hatching the insignificant data as suggested, but for figure 4 and 6, because there are less significant data than insignificant data, especially for GSH classification, it does not contribute to improve the clarify, instead, it may highlight the insignificant data. Thus, after thinking twice, we do not change figure 4 and figure 6 in the revised manuscript. For figure 8, as suggested, we revised the figures to hatch the

insignificant data. We tried different hatches, i.e. slashes, cross signs, and dots, and finally choose dots for insignificant data, as it is better to highlight the shades and arrows, and make it able to distinguish this figure from other figures that hatch significant data with slashes and cross signs. We hope this makes the figure clear now. Thanks a lot for the suggestion.

The figures with the little dots and hashes that are one way for insignificant in some and the other way in others is not good either.

*Figure 8 – the winds plus the hatching make this figure incredibly difficult to read.*

To make it clearer, we revised the figure to hatch the insignificant data as suggested above, and changed the slashes and cross signs to dots to avoid covering shades and arrows.

*Figure 9 – given that the stratospheric aerosol fields were \*zonally averaged\* for PMIP3/CMIP5, I am not sure that a map-view figure is needed.*

Something is really odd with this figure. Why is the whole of GNH hashed and GSH hashed hardly at all? Why is there a line right across 5N in GSH? Is OSR is really in excess of 20W/m2 throughout the whole tropics in GNH but near 0 for GSH? There could be a problem with the diagnostics. This may be a sampling issue as at the top, or it may be a fundamental issue with the diagnostics or ??? But, the present plotting doesn't really make sense. If there authors DO decide to keep their NH SH classification, there needs to be some attempt to scale the vastly different forcing applied to each such that they are comparable when lined up on figures like this.

Yes, the volcanic forcing is zonally averaged, but as shown in the figure, the clear-sky OSR is not exactly the same as the forcing, which is not totally identical at the same latitude, especially for the GSH classification, besides, the map-view figure is useful to be compared with the full-sky OSR map-view in figure 10. Considering these, we think it helps to keep the map-view figure.

*Figure 10 – the figure label here for the top row looks identical to figure 9 but the figures are different (put why in the caption). C is not a good shorthand for percent cloudiness, 'cloud total' or similar is better.*

The y-axis labels are clearly an improvement.

Figure 9 is outgoing shortwave radiation in clear sky, while figure 10 is in full sky, we wrote the difference in the figure caption. Considering the comment, we think it's better to make it clearer in the figure itself, so we revised the y-label in figure 9 and figure 10 to Clear-sky OSR and Full-sky OSR, specifically. We used C to make it shorter in the figure caption, but clarity is more important, considering the comment, we replaced the figure caption directly with "Cloud area fraction" in the revised figure.

---

## Author Response (AR3)

**Response to editor's comments**
**"Mechanisms of hydrological responses to volcanic eruptions in the Asian monsoon and westerlies-dominated subregions" by Zhihong Zhuo et al.**

We are very grateful to Editor Allegra N. LeGrande for your efforts and time. We have considered the comments thoroughly and carefully read and compared this study and studies you mentioned, i.e. Colose et al. (2016) and Zanchettin et al. (2022). Considering differences in main focuses of different studies, we did minor revisions in the manuscript, and give our detailed reasons here. The list of the editor's comments (in red) as well as our responses (in black) are listed below. The revised texts are shown in blue.

Upon re-reading this manuscript and puzzling over what seems like odd figures, I realize that Figure 2 is really rather inconsistent with Colose et al 2016 Figure 1. This becomes important for this whole paper because in the present Zhou analysis, there is really only 1 example of a 'large' southern hemisphere eruption. I don't really understand why this is, so I replotted the volcanic forcings for GISS (these are in AOD space, not Tg loading space, but the former is scaled to apply forcing to climate models). If we zoom in on the 1450's, we see NH (20-90N), TR (20S-20N), and SH(90S-20S) look like this… so why is it classified as GNH?

The criteria for classifying volcanic events is different between Colose et al. (2016) and this study. Colose et al. (2016) classified volcanic events into three categories: $ASYMM_{NH}$, $ASYMM_{SH}$ and SYMM, based on a "25 % difference in aerosol loading between hemispheres" criteria. As can be seen from the category name, it emphasized the symmetric and asymmetric forcing. However, in this study, the classifications, i.e. GNH and GSH, emphasize the aerosol injection in specific hemispheres, as was defined in the manuscript "Following Zhuo et al. (2020), we pick out volcanic events in 1300-1850 CE that have larger northern hemisphere volcanic aerosol injection (NHVAI) than that of the 1991 Mount Pinatubo eruption (17 Tg $SO_2$ based on the GRA volcanic forcing index) as GNH classification. To explore inversed hydrological impacts of interhemispherically asymmetric VAI, another classification, with volcanic events in 1300-1850 CE that only have southern hemisphere volcanic aerosol injection (SHVAI), is constructed as the GSH classification.". To meet the criteria, based on the Gao et al. (2008) volcanic forcing reconstruction, there is only one event with large SH aerosol injection than Pinatubo magnitude. Thus the criteria for the aerosol magnitude of the GNH and GSH classification is also different. As noted in the manuscript, considering the limited aerosol magnitude, but the same number of 12 volcanic events in the GSH classification, GSH classification mainly serve as a reference for the GNH classification, to show the difference between with and without NH aerosol injection.

For the 1452 event, we knew that, based on Gao et al. (2008) volcanic forcing reconstruction, it has 44.6 Tg and 92.9 Tg aerosol loading in the NH and SH, respectively. But climate in the NH is largely affected by the large aerosol injection in the NH, especially for regional climate responses. Thus, we still included it, as it meets the classification criteria with NHVAI much larger than that of the 1991 Mount Pinatubo eruption. We know that uncertainties exist from studies based on volcanic classifications, but also other sources. Zhuo et al. (2020) had a whole section discussing sources of uncertainty. We know that different volcanic events have different impacts, especially regional impacts, which largely depends on the source parameters of the volcanic eruptions and pre-eruption initial conditions of the climate system. These are important questions to be further explored and answered by the research community, and is also a focus of researches we are doing now.

For my previous comments…
Above, the authors do not mention the VolMIP results that specifically explore the impact of the initial ENSO state on the simulated response. https://gmd.copernicus.org/articles/15/2265/2022/

It is too important and relevant to the discussion. I disagree that the relatively small ensemble used here (and in the previous paper) is an adequate stand-in for ENSO discussion. Its needs its own section plus context woven throughout the text. The models used here DO INDEED appear in Davide's analysis… except that there was intentional sampling across a variety of initial conditions. It should be pretty straight-forward to 'look up' the range of responses, the implications for precip, convergence, temperature, etc. and carry on. This paper really does haveinsights into how initial condition projects onto temperature, heat fluxes, etc. And it is a MUCH larger sampling of volcanic forcing which is rather ephemeral, and sometimes hard to tease out beyond the background 'noise'/variability.

Sorry, the "Zanchettin et al., 2020" was a typo in the previous "response to editor" letter. In the previous revised manuscript, it was written as "...its potential future phases with improved protocol addressing the pre-eruption ENSO state (Zanchettin et al., 2022) can be valuable resource to investigate these questions.", which refers to the same VolMIP paper as suggested by the editor.

In Zhuo et al. (2020), it was said that "A limited number of ensemble members might also introduce uncertainty in the model results", but it didn't mean it is a stand-in for ENSO discussion. There is a whole paragraph in Zhuo et al. (2020) discussing ENSO and hydrological impacts. As investigated, the hydrological pattern remain largely unchanged after regressing out ENSO. Thus ENSO can not expain different hydrological responses between the westerlies and monsoon-dominated subregions of Asian. To understand this is the main focus of this study, instead of the connection between ENSO and hydrological impacts itself. Thus, we didn't discuss ENSO in order to avoid redundance.

Yes, part of the models used in this study appear in Davide's response, but this study used the CMIP5 model data, while models in Davide's study are in their CMIP6 version. As stated in Zelinka et al. (2020), "climate sensitivity is larger on average in CMIP6 (1.8–5.6 K) than in CMIP5 ( 2.1-4.7 K) due mostly to a stronger positive low cloud feedback, but recent consensus places it likely within 1.5–4.5 K". We evaluated hydrological responses in the AMR in CMIP5 before investigating the mechanisms. Definitely, it's valuable to further investigate the mechanisms with both CMIP6 and VolMIP data after evaluating the applicability of CMIP6 and VolMIP for regional studies. This was stated in the manuscript, "Future studies with PMIP4/CMIP6 data (Jungclaus et al., 2017) with updated volcanic forcing reconstruction (Toohey and Sigl, 2017) can contribute to better understanding on this topic." and "The Model Intercomparison Project on the climatic response to Volcanic forcing (VolMIP, (Zanchettin et al., 2016) and its potential future phases with improved protocol addressing the pre-eruption ENSO state (Zanchettin et al., 2022) can be valuable resource to investigate these questions."

We completely agree that Davide's study is very important, and VolMIP simulations are very useful for understanding the impact of volcanic eruptions that beyond the background noise/variability. When this study was conducted, the VolMIP data was not available yet. Actually, before conducting this study, I considered to use data from VolMIP and visited Claudia Timmreck at the Max Planck institute, but was told that it still takes time to conduct VolMIP simulation at that time. We didn't mention the detailed results of Davide's study on the impact of the initial ENSO state on the simulated response, because we didn't focus on understanding the initial state of ENSO on the hydrological impacts of volcanic eruptions. This is a very important and comprehensive research question that needs in depth investigation, which, actually, is the main focus of the study we are doing now. We don't want to mess it up with a brief discussion in this study. But we would like to emphasize the important of this topic referencing Davide's study, thus we revised the discussion as shown in line 338-349 in the revised manuscript:

The disagreement may result from different initial state of ENSO used in different studies. Through analyzing the model output from the Model Intercomparison Project on the climatic response to Volcanic forcing (VolMIP, Zanchettin et al., 2016), Zanchettin et al. (2022) found that the pre-condition of ENSO impacts temperature and precipitation responses after volcanic eruptions. This points out the importance of investigating the dependency of post-eruption ENSO and hydrological responses to preeruption initial conditions. More studies are also needed to understand the ENSO and hydrological responses to SH volcanic eruptions. Besides, the interaction between post-eruption ENSO and monsoon precipitation varies in different monsoon regions. A weakened African monsoon due to post-eruption cooling in Africa leads to the El Nino response after tropical eruptions (Khodri et al., 2017), but a more frequent occurrence of El Niño in the first boreal winter after eruptions lead to an enhanced EASM (Liu et al., 2022). The interaction among ENSO, monsoon and volcanic eruptions remains unclear. The VolMIP (Zanchettin et al., 2016) and its potential future phases with improved protocol addressing the pre-eruption ENSO state (Zanchettin et al., 2022) can be valuable resource to investigate these questions.

Here is a place where you could add an extra sentence wrt ENSO.
This paragraph focused on the mechanisms explored by previous studies and the needed studies focusing on regional scales of the AMR. We acknowledge that ENSO is related to the hydrological response, but it can not explain regional differences of the hydrological patterns in the AMR. Zhuo et al. (2020) investigated this and showed that consistent with Iles et al. (2013) and Iles and Hegerl (2014), regressing out the effect of ENSO only results in a lower amplitude response, but the temporal and spatial patterns remain largely unchanged.". Thus, it is not the ENSO that lead to different hydrological impacts between the RWA and RDA, whereas, the aim of this study is to explore the mechanism of the different regional hydrological responses. Thus, we do not think it fits to add a sentence wrt ENSO here, but as suggested previously by the editor, and combined with the comment above, we think it's important to discuss ENSO-related uncertainties and unresolved questions. Thus, we added a sentence wrt ENSO and Davide's study in the concluding remarks, as shown in the answer to the above comment.

The figures with the little dots and hashes that are one way for insignificant in some and the other way in others is not good either.
Like every coin has two sides, different figures also have different advantages and disadvantages. As answered previously, we tried different ways and finally used this. Because the others with slashes and plus signs show the significant results, then with dots for the insignificant results. These can show the main results. And with different signs, it can exactly show the change from highlighting significant data to insignificant data, and the dots won't affect the view of wind arrows. We think this works after thinking twice.

Something is really odd with this figure. Why is the whole of GNH hashed and GSH hashed hardly at all? Why is there a line right across 5N in GSH? Is OSR is really in excess of 20W/m2 throughout the whole tropics in GNH but near 0 for GSH? There could be a problem with the diagnostics. This may be a sampling issue as at the top, or it may be a fundamental issue with the diagnostics or ??? But, the present plotting doesn't really make sense. If there authors DO decide to keep their NH SH classification, there needs to be some attempt to scale the vastly different forcing applied to each such that they are comparable when lined up on figures like this.
This figure exactly reflects the different aerosol injection magnitude in the GNH and GSH classification. Based on the criteria, all the events included in the GNH classification have aerosol injection magnitude than that of the 1991 Pinatubo eruption, the included events have aerosol injections that are 1~5.5 (average 2.2) times larger than Pinatubo eruption, thus it largely reduces the clear sky shortwave radiation by reflecting the incoming solar radiation. Zhuo et al. (2021) simulated Pinatubo strength eruption with MPI-ESM, which showed up to 8~9 $W/m^2$ reduction of global mean clear sky shortwave radiation. 20 $W/m^2$ of reduction is reasonable, considering the 1~5.5 (average 2.2) times of injected aerosols. For GSH, there is no NH aerosol injection, and the aerosol injections in the SH are small, it's reasonable to have limited response, especially in the NH. The line across 5N is not a

diagnostic problem, but relatively uniform responses to zonally averaged forcing which are in the same colorbar scale range. If we use a smaller scale with a lot of levels, as shown in the figure below, slight differences emerge across 5N.

[Figure]

However, the scale is used to emphasize the GNH classification and the RWA and RDA, instead of the GSH and tropical regions, and aimed to show the difference between the GNH and GSH classification.

The main focus of this study is on the mechanism exploration, instead of the classification and related responses, as stated in the manuscript and explained above, GSH has limited magnitude, we include this classification, because it has the same 12 events included in the classification, which serves as a reference to the GNH classification, and show the contrast between with and without NH aerosol magnitude. The aerosol magnitude in the GSH classification does not affect the main focus of the study.

The y-axis labels are clearly an improvement.
Thanks.

---

## Author Response (AR4)

**Response to editor's comments**
**"Mechanisms of hydrological responses to volcanic eruptions in the Asian monsoon and westerlies-dominated subregions" by Zhihong Zhuo et al.**

Editor's comment:
You might consider, whether the two anonymous reviewers shouldn't be thanked for their contributions?

Dear Co-chief editor Marit-Solveig Seidenkrantz,

Thank you very much for taking over the responsibility to act as editor of our manuscript and nice notes. We are glad that the manuscript is finally going to be accepted for publication. Thank you for your kind reminder on adding a thanks to reviewers. Yes, we definitely would like to thank the reviewers for their contributions. Their constrictive comments helped a lot on improving the quality of our manuscript. As suggested, we added following text "We would like to thank the reviewers, Fei Liu and another anonymous reviewer, for their helpful and constructive comments." in the Acknowledgments (marked with yellow background in the file with tracked changes).

We know that it's a rare case regarding the unusual delay of the process. It's not the problem of the journal itself. This won't affect our potential future submission to the journal. Thank you for your nice notes again.

Best regards,
Zhihong, Ingo and Ulrich